# Natural soap is clinically effective and less toxic and more biodegradable in aquatic organisms and human skin cells than synthetic detergents

**Takahide Kanyama[1], Akihiro Masunaga[1], Takayoshi Kawahara[1,2], Hayato Morita[3], Sadanori Akita[4,5]***

1 Research and Development Department, Shabondama Soap Co., Ltd., Kitakyushu, Fukuoka, Japan, 2 Faculty of Environmental Engineering, The University of Kitakyushu, Kitakyushu, Fukuoka, Japan, 3 Shabondama Soap Co., Ltd., Kitakyushu, Fukuoka, Japan, 4 Fukushima Medical University, Fukushima, Japan, 5 Tamaki Aozora Hospital, Tokushima, Japan

* akitas@hf.rim.or.jp

## Abstract

In the era of COVID-19, concerns about and consumption of soaps and detergents have increased. The environmental effects, along with their direct impacts on the human body, are being simultaneously considered to ensure safety and support healthy living. Natural soap compounds are considered readily biodegradable and unlikely to produce hazardous waste, while artificial detergents are composed of synthetic surfactants, plasticizers, binders, and additives. This study aimed to investigate representative natural soap compounds consisting of fatty acid salts and compare them with synthetic detergents, such as sodium dodecylbenzene sulfonate (SDB) and sodium lauryl sulfate (SLS). Environmental assays recommended by the OECD, as well as human keratinocyte assays for toxicity and biodegradability, were utilized. The major components of natural soap were found to be less toxic and more biodegradable in aquatic environments—assessed using algae, crustaceans, and fish—compared to synthetic detergents. Additionally, in the human keratinocyte assay, natural soap compounds were significantly less toxic and demonstrated higher viability than SLS after a 48 h culture and a 5 min exposure. The half-maximal inhibitory concentration ($IC_{50}$) obtained from the viability assay revealed values of 7.82 mM for potassium laurate (C12K), 7.56 mM for potassium oleate (C18:1K), and 0.604 mM for SLS. Therefore, natural soap appears to be valuable due to its lower toxicity, greater biodegradability in aquatic environments, enhanced safety for human cells, and potential efficiency in clinical applications.

## Introduction

Soap and its major component, fatty acid salts, are known to exhibit oral antibacterial activity [1], inactivate human and avian influenza viruses in vitro [2], demonstrate

**Data availability statement:** "The relevant data are within the paper and can be accessed at the following link: (https://doi.org/10.6084/m9.figshare.28746467.v2)."

**Funding:** This study was partly supported by non-restricted funds from the funder (Shabondama Soap Co., Ltd.) and we did not receive any other grant or external funding. The funding source provided support in the form of salaries for TKan, AM and TKawa. HM is the board of the funding source. However, he had no role in the study design, data collection and analysis, decision to publish, or preparation of the manuscript. The specific roles of these authors are articulated in the 'author contributions' section.

**Competing interests:** The funding source (Shabondama Soap Co., Ltd.) provided support in the form of salaries for TKan, AM and TKawa. This does not alter our adherence to PLOS ONE policies on sharing data and materials.

**Abbreviations:** SLS, sodium lauryl sulfate; SDB, sodium dodecylbenzene sulfonate; LDH, lactate dehydrogenase; C12Na, sodium laurate; C14Na, sodium myristate; C18Na, sodium stearate; C18:1Na, sodium oleate; C12K, potassium laurate; C14K, potassium myristate; C18K, potassium stearate; C18:1K, potassium oleate; OECD, Organisation for Economic Cooperation and Development; $EC_{50}$, 50% effect concentration; $LC_{50}$, 50% lethal concentration; NOEC, no observed effect concentrations; DOC, dissolved organic carbon; MITI, Ministry of International Trade and Industry; HBSS, Hanks' Balanced Salts Solution; $IC_{50}$, half maximal inhibitory concentration; ANOVA, analysis of variance; WST, water-soluble tetrazolium; PRTR, Pollutant Release and Transfer Register; LAS, linear alkylbenzene sulfonates.

bactericidal properties, remove *Staphylococcus aureus* biofilm, and exhibit reduced cytotoxicity in fibroblasts and keratinocytes [3]. They also improve human dermal fibroblast viability, reduce cytotoxicity, and accelerate the activity of human epidermal keratinocytes in vitro, as well as promote healing in human chronic wounds in vivo [4].

There are two main types of soap: one is a natural-based compound derived from plant and animal oils and fats, and the other is synthetic-based, made from synthetic surfactants, plasticizers, and additives, often formulated from fats, petroleum, and oil-related products [5].

In healthcare environments, maintaining elevated levels of hygiene is highly recommended through handwashing with chlorhexidine sanitizer and natural soap, which have been shown to be as effective as synthetic soaps in a randomized crossover study. Moreover, handwashing with natural soap in the operating room was rated as more beneficial than synthetic soap, based on subjective scores from questionnaires evaluating foaming, quality, and longevity [6].

In the era of COVID-19, the frequency of handwashing and household soap consumption has increased [7]. Soaps are recommended for maintaining hygiene and preventing disease or pathogen transmission [8,9]; however, more frequent handwashing can result in a higher incidence of hand contact dermatitis, leading to irritation and dryness [10].

Since soaps and synthetic detergents are used in healthcare units, hospitals, and clinics for personal hygiene and cleaning applications, the wastewater flow near these facilities is a significant concern due to the higher concentration of surfactants. These surfactants can undergo both aerobic and anaerobic degradation in aquatic and terrestrial environments [11]. The toxic components of surfactants may affect living organisms, including bacteria and mammals, and can even alter non-living environments [12].

Furthermore, surfactants and their degradation products accumulate in wastewater treatment plants, altering the physicochemical properties of surface waters, wastewater residues, and soils [13]. Notably, petrochemical-based soaps and alcohol-based hand rubs are sometimes compared to more eco-friendly alternatives derived from plants and microbes, even though they are made from fossil fuels [14]. Recently, the COVID-19 pandemic highlighted the negative impacts of human activities on the ecosystem. For example, water pollution can disrupt aquatic flora and fauna, increase wastewater, and contaminate lakes, rivers, and oceans [15]. Importantly, natural soap does not produce toxic waste or byproducts and requires less energy to produce [16]. These soaps consist of several fatty acid salts, which are composed of various long-chain carboxylic acids combined with alkali metals, usually potassium or sodium [3]. The active ingredients in natural soaps are made solely from fatty acid salts [4]. In contrast, synthetic soap is made from petroleum-based ingredients, such as sodium lauryl sulfate (SLS) and sodium dodecylbenzene sulfonate (SDB), which are toxic and less biodegradable [17–19]. Detergents claiming to contain natural and petrochemical-free ingredients have been found to contain petroleum-derived

components, as determined by radiocarbon isotope (carbon-14) analysis. This finding may have implications for product safety and skin benefits [20].

Even though the addition of newly synthesized Evening Primrose amidopropyl sulfobetaine to surfactants containing SLS may potentially improve safety in SLS-based liquid washing soap, there is limited data on natural soap or its ingredients [21].

We aimed to investigate the toxicity of natural soap components in aquatic organisms, such as algae, crustaceans, and fish [22], as well as their biodegradability. We also tested cytotoxicity in human keratinocytes by measuring lactate dehydrogenase (LDH) release as an indicator of plasma membrane damage [23] and assessed human accessibility for use, as confirmed in surgical handwashing [6]. Direct toxicity and biodegradability in the environment, along with effects on keratinocytes, may contribute to one health and exposome [24].

## Materials and methods

### 1.1. Reagents and fatty acid salts

Sodium laurate (C12Na), sodium myristate (C14Na), sodium oleate (C18:1Na), potassium oleate (C18:1K), and SDB were purchased from Tokyo Chemical Industry Co., Ltd. (Tokyo, Japan). Potassium laurate (C12K), potassium myristate (C14K), sodium oleate (C18:1Na), and SLS were purchased from FUJIFILM Wako Pure Chemical Corporation (Osaka, Japan). Sodium stearate (C18Na) and potassium stearate (C18K) were purchased from JUNSEI CHEMICAL Co., Ltd. (Tokyo, Japan). Bar soap and liquid soap made by saponification method were provided by Shabondama Soap Co., Ltd. (Fukuoka, Japan)(Fig 1).

### 1.2. Toxicity effects on aquatic organisms

Toxicity effects of soap components were conducted on three species of aquatic organisms, algae, crustaceans, and fish according to OECD (Organisation for Economic Cooperation and Development) test guidelines to calculate 50% effect concentration ($EC_{50}$), 50% lethal concentration ($LC_{50}$), or no observed effect concentrations (NOEC). The details are described below.

**1.2.1. Evaluation of algae ($EC_{50}$, NOEC).** Following the OECD 201 [25], The Freshwater Algae and Cyanobacteria Growth Inhibition Test was conducted to assess the effects of a test substance on the growth of freshwater microalgae and related organisms. Exponentially growing test organisms were exposed to the test substance at a range of concentrations for 72 h. In this test, five different concentrations arranged in a geometric series with a factor of 2.0–2.2 were prepared in OECD medium. Following OECD 201 recommendations, for each of the five concentrations, three

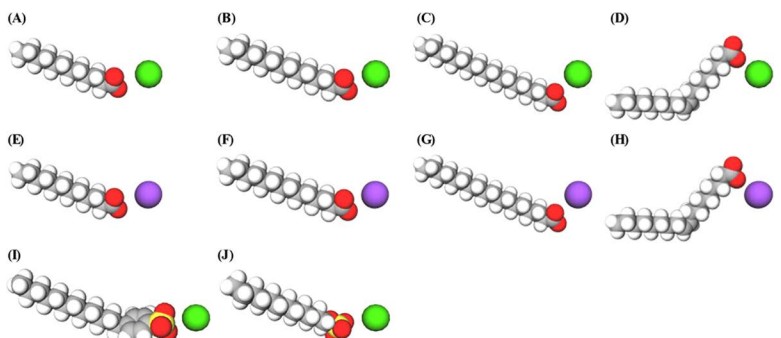

**Fig 1. Structures of the fatty acid salts and synthetic detergents. (A)** C12Na; **(B)** C14Na; **(C)** C18Na; **(D)** C18:1Na; **(E)** C12K; **(F)** C14K; **(G)** C18K; **(H)** C18:1K; **(I)** SDB; and **(J)** SLS. Carbon: gray, hydrogen: white, oxygen: red, sulfur: yellow, potassium ions: purple, sodium ions: green.

replicates were used, while six replicates were included for the control group. The initial algal cell concentration was set at $1 \times 10^4$ cells/mL, and *Pseudokirchneriella subcapitata (Raphidocelis subcapitata)* was evaluated after being exposed to each concentration for 72 h.

**1.2.2. Evaluation of crustacean (EC$_{50}$, NOEC).** Following OECD Test Guideline 202 [25], the *Daphnia sp.* Acute Immobilization Test was conducted to assess the effects of a test substance on the immobilization of daphnids. Young daphnids, less than 24 h old at the start of the test, were exposed to the test substance at a range of concentrations for a period of 48 h. In this test, five different concentrations arranged in a geometric series with a factor of 2.0 were prepared in M7 medium. Twenty *Daphnia magna* were divided into four groups of five each. *D. magna* were evaluated after being exposed to each concentration for 48 h.

Following OECD Test Guideline 211 [25], the *Daphnia magna* Reproduction Test was conducted to assess the effects of a test substance on the reproductive output of *D. magna*. Young female *D. magna*, less than 24 h old at the start of the test, were exposed to the test substance at a range of concentrations for 21 days. In this test, five different concentrations arranged in a geometric series with a factor of 1.5–1.6 were prepared in dechlorinated tap water. Ten *D. magna* were maintained individually, one per test container, in a semi-static system. *D. magna* were evaluated after being exposed to each concentration for 21 days.

**1.2.3. Evaluation of fish (LC$_{50}$, EC$_{50}$, NOEC).**

**1.2.3.1 *Oryzias latipes* maintenance** In the fish test, *O. latipes* were tested at an ISO 9001 certified institution. *O. latipes* were maintained in aquariums (20 fish/ 5 L) at room temperature (22–26°C) during the test period. Water was changed once a day and feed was provided once a day. The endpoint for LC$_{50}$ was the observation of mortality, and endpoint for EC$_{50}$ was the observation of mortality and abnormal behavior such as drifting on the bottom. After testing, fish were placed at 3°C for sleep and then euthanized at -18°C.

**1.2.3.2 *O. latipes* evaluation** Following OECD Test Guideline 204 [26], the Fish Prolonged Toxicity Test was conducted to assess the effects of a test substance on lethal and other observable effects in fish. In this test, five different concentrations arranged in a geometric series with a factor of 2.0–2.8 were prepared using well water. Twenty *O. latipes* were evaluated after being exposed to each concentration for 14 days under semi-static conditions, which were more stringent than those outlined in OECD Test Guideline 203 for Fish Acute Toxicity Testing.

## 1.3. Biodegradability tests

The biodegradability test of soap components was conducted by two methods according to OECD test guidelines. The details are described below.

**1.3.1. OECD 301A.** The DOC Die-Away Test was conducted according to OECD test guideline 301A [27]. A mixture of mineral medium 1 L, the test substance 2 ml, and activated sludge 2 ml were incubated for 28 days at 22±2°C. The dissolved organic carbon (DOC) in the test solution after 0 days and 28 days was measured, and the biodegradation rate was calculated using the Equation below. The index of easy degradability is a biodegradation rate of 70% or more after 28 days.

$$Degradation\ (\%) = 100 \times \left(1 - \frac{(DOC\ at\ 28\ days) - (DOC_{blank}\ at\ 28\ days)}{(DOC\ at\ 0\ days) - (DOC_{blank}\ at\ 0\ days)}\right)$$

(1)

where $DOC_{blank}$ is the DOC of the blank area.

**1.3.2. OECD 301C.** A modified Ministry of International Trade and Industry (MITI) Test was conducted according to OECD test guideline 301C. A mixture of mineral medium 290 ml, the test substance 30 mg, and activated sludge 10 ml were incubated for 28 days at 25±1°C. After incubation, the BOD, Biochemical oxygen demand, was measured and the biodegradation rate was calculated using the Equation below. The index of easy degradability is a biodegradation rate of 60% or more after 28 days.

$$Degradation\ (\%) = 100 \times \frac{BOD - BOD_{blank}}{ThOD} \tag{2}$$

Where $BOD_{blank}$ is the BOD of the blank area and $ThOD$ is the theoretical oxygen demand. The biological half-life of each test sample, $t_{1/2}$, was obtained by the following equation,

$$BOD_r\ (\%) = ThOD \times e^{-k_1 t} \tag{3}$$

$$t_{\frac{1}{2}} = \frac{ln\ 2}{k_1} \tag{4}$$

Where $BOD_r$ is the BOD remaining at day $t$ ($ThOD$-$BOD$) and $k_1$ is the degradation rate constant. The value of $k_1$ was evaluated by plotting $ln\left(\frac{BOD_r}{ThOD}\right)$ vs. day $t$ trading linear approximation.

## 1.4. Human keratinocyte cytotoxic assay

Fatty acid potassium salts, C12K, C18:1K, and the synthetic surfactant, SLS, were used in the cytotoxicity assays. Hanks' Balanced Salts Solution (HBSS) (+) buffer, containing Ca and Mg, was used to produce the fatty acid solutions. Both fatty acid potassium salts (C12K and C18:1K; 0.01, 0.0312, 0.1, 0.312, 0.5, 1, 3.12, 5, 10, 31.2 mM) were prepared by mixing the relevant fatty acid (C12 and C18:1, respectively) with KOH that had been solubilized with HBSS (+) at 80°C. Then, the pH of each fatty acid salt was adjusted to 10.4 by adding KOH. SLS (the same molality as fatty acid salts) was prepared by diluting them with HBSS (+). This solution had a final pH value of 7.7.

   1.4.1. **Cell culture.** A primary human keratinocyte cell line, NHEK-Ad cells (Lonza Japan Ltd., Tokyo, Japan) were cultured in KBM-Gold medium (Lonza Japan). In total, $1 \times 10^4$ cells were added to each well of a 96-well Nunc MicroWell microplate (Thermo Fisher Scientific) before being cultured for 48 h in a humidified atmosphere containing 5% $CO_2$ at 37°C.

   1.4.2. **Cytotoxic assay.** LDH leakage and cell viability were evaluated as indices of cytotoxicity. NHEK-Ad cells were cultured for 48 h and treated with 100 µL/well of C12K, C18:1K, or SLS for 5 min at room temperature. HBSS (+)-treated cells were used as the control. Then, the solution was collected from each well and used for the LDH leakage assay. NHEK-Ad cells were washed three times with the culture medium and then their viabilities were evaluated (Fig 2). The half maximal inhibitory concentration (IC$_{50}$), values of each solution for NHEK-Ad cells were calculated from cell viability using the next equation,

$$IC_{50}\ [mM] = 10^{\left(\frac{50-y_B}{y_A-y_B}\ log\left(\frac{m_A}{m_B}\right)+log(m_B)\right)} \tag{5}$$

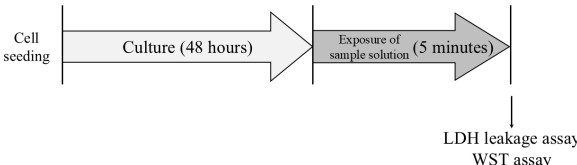

**Fig 2. Cytotoxicity test for the NHEK-Ad cells.** NHEK-Ad cells that had been cultured in KBM-Gold medium for 48 h were treated with 100 µL/well of each sample for 5 min. The solution was collected from each well and used for the LDH leakage assay. After that, their viability was assessed.

where, $m_A$ and $m_B$ were the higher and lower concentration between 50% cell viability, respectively, and $y_A$ and $y_B$ were the cell viability at the molality $m_A$ and $m_B$, respectively. LDH leakages and cell viabilities were assessed in each well using the cytotoxicity LDH assay kit-WST (Dojindo Laboratories, Kumamoto, Japan) and cell counting kit-8 (Dojindo Laboratories), respectively (Fig 2).

### 1.5. Questionnaires for clinical use

Details of the study protocol were explained to all 50 patients with skin conditions or scars, including 16 with acne, 10 with scars, 10 post-LASER treatments, 5 with burn, 4 with surgery, 1 with inflammatory, 1 with plaque LASER, 1 with herpes, 1 with finger nail injury, and 1 with acne and post-LASER treatments. Informed consent was obtained from all participants. The inclusion and exclusion criteria are listed in Table 1. Some parts in the inclusion criteria were based on the "Classification of acne severity" [28]. The study was approved by the Ethical Review Committee of Tamaki Aozora Hospital (Institutional Review Board, IRB number 005–2024). All written informed consents were taken before starting the study under declaration of Helsinki. Patients were asked a series of questions after washing the affected area with soap, and the degree of each response was evaluated on a four-point scale, ranging from 1 = "feel bad" to 4 = "feel good." The questionnaires partially mirrored those used with "healthy volunteers" in previous research [6].

1. How itchy is the skin after washing with soap?

2. How sticky is the skin after washing with soap?

3. How moist is the skin after washing with soap?

4. How dry is the skin after washing with soap?

5. How tight is the skin after washing with soap?

### Statistic

The data of cytotoxic assay and questionnaires were expressed mean ± standard deviation. For statistical analysis, the data of toxicity effects on algae and crustacean were analyzed in R software and Ecotox-Statics ver.3.01.01, respectively. In the evaluation of algae, the linear regression analysis was used for $EC_{50}$ and the one-way analysis of variance (ANOVA) was used for NOEC. In the evaluation of crustacean, probit analysis was used for $EC_{50}$ and Dunn's test was

**Table 1. Criteria for patients to participate in the present study.**

| Inclusion criteria | 1. Written informed consent to participate in the study |
| --- | --- |
| | 2. Patients with skin diseases such as acne vulgaris, burns, surgical scars, laser scars, inflammation (post-operative and post laser exposure) |
| | 3. For patients with acne vulgaris of the face, mild, moderate, or severe symptoms according to the "Classification of acne severity"※[28], patients with acne vulgaris on other parts of the face or other diseases with symptoms judged by the physician to be of similar severity.<br>[Classification of acne severity]<br>Mild: 5 or fewer cases of inflammatory eruptions in the half face<br>Moderate: more than 6 but less than 20 cases of inflammatory eruptions in the half face<br>Severe: more than 21 but less than 50 cases of inflammatory eruptions in the half face<br>Very severe: 51 or more cases of inflammatory eruptions in the half face |
| Exclusion criteria | 1. Patients using oral or injectable steroid containing products |
| | 2. Patients using other cleaning agents to clean the affected area |
| | 3. Patients receiving antibacterial, antibiotic, or antiviral agents on affected area |

used for NOEC. In the evaluation of fish, binomial distribution was used. The data of LDH leakage and cell viability were analyzed by the one-way ANOVA. p-Values <0.05 were considered significant.

## Results

### 2.1 Reagents and fatty acids

Several natural soap compounds made of fatty acids with potassium and sodium salts were tested, and the synthetic detergents SDB and SLS were used for comparison (Fig 1).

### 2.2 Toxicity effects on aquatic organisms

**2.2.1 Evaluation of algae (EC$_{50}$, NOEC).** Among compounds of natural soap, the median effective concentrations at 72 h (72h-EC$_{50}$) exhibited by *P. subcapitata* in C12Na, C12K, C14Na, C14K, C18Na, C18K, C18:1Na, and C18:1K revealed 96, 50, 127, 147, 77, 80, 115, and 106 mg/l, respectively.

Synthetic detergent compounds SDB and SLS demonstrated 39 and 24 mg/l, respectively.

The NOEC exhibited by *P. subcapitata* in C12Na, C12K, C14Na, C14K, C18Na, C18K, C18:1Na, and C18:1K revealed 39, 16, 74, 94, 38, 32, 65, and 39 mg/l, respectively.

Synthetic detergent compounds SDB and SLS demonstrated 36 and 5.8 mg/l, respectively.

A solid (i.e., bar) and liquid soap were also tested in this experiment, which is composed of multiple fatty acid salts, and they demonstrated 115 and 148 mg/l for 72h-EC$_{50}$ and 65 and 111 mg/l for NOEC, respectively (Table 2).

**2.2.2 Evaluation of crustaceans (EC$_{50}$, NOEC).** Among the compounds of natural soap, 48h-EC$_{50}$ exhibited by *D. magna* in C12Na, C12K, C14Na, C14K, C18Na, C18K, C18:1Na, and C18:1K revealed 64, 71, 66, 40, 59, 17, 40, and 91 mg/l, respectively.

Synthetic detergent compounds SDB and SLS demonstrated 27 and 13 mg/l, respectively.

For referral, the NOEC exhibited by *D. magna* in C18:1Na and C18:1K revealed 0.322 and 0.413 mg/l, respectively. For the soaps composed of multiple fatty acid salts, they demonstrated 72 and 84 mg/l for 48h-EC$_{50}$ in the bar and liquid soap, respectively, and 3.59 mg/l for NOEC in bar (solid) soap (Table 2).

**2.2.3 Evaluation of fish (LC$_{50}$, EC$_{50}$, NOEC).** Among the compounds of natural soap, the median lethal concentrations at 7 days (7d-LC$_{50}$) exhibited by *O. latipes* in C18:1Na and C18:1K showed 429 and 448 mg/l, respectively,

**Table 2. Toxicity of fatty acid salts, soaps and synthetic detergents.**

| No. | Sample | *P. subcapitata* | | *D. magna* | | *O. latipes* | | |
|---|---|---|---|---|---|---|---|---|
| | | OECD 201 | | OECD 202 | OECD 211 | OECD 204 | | |
| | | 72h-EC$_{50}$ (mg/l) | NOEC (mg/l) | 48h-EC$_{50}$ (mg/l) | NOEC (mg/l) | 7d-LC$_{50}$ (mg/l) | 14d-LC$_{50}$, 14d-EC$_{50}$ (mg/l) | NOEC (mg/l) |
| 1 | C12Na | 96 | 39 | 64 | – | – | – | – |
| 2 | C12K | 50 | 16 | 71 | – | – | – | – |
| 3 | C14Na | 127 | 74 | 66 | – | – | – | – |
| 4 | C14K | 147 | 94 | 40 | – | – | – | – |
| 5 | C18Na | 77 | 38 | 59 | – | – | – | – |
| 6 | C18K | 80 | 32 | 17 | – | – | – | – |
| 7 | C18:1Na | 115 | 65 | 40 | 0.322 | 429 | 410 | 98 |
| 8 | C18:1K | 106 | 39 | 91 | 0.413 | 448 | 438 | 97 |
| 9 | Bar soap | 115 | 65 | 72 | 3.59 | 158 | 158 | 101 |
| 10 | Liquid soap | 148 | 111 | 84 | – | 1,003 | 902 | 284 |
| 11 | SDB | 39 | 36 | 27 | – | 4.7 - 6.6 ※ [35, 36] | | – |
| 12 | SLS | 24 | 5.8 | 13 | – | 2.8 - 30.5 ※ [37-40] | | – |

while those 14d-LC$_{50}$ (EC$_{50}$) exhibited by *O. latipes* in C18:1Na and C18:1K showed 410 and 438 mg/l, respectively. NOEC exhibited by *O. latipes* in C18:1Na and C18:1K showed 98 and 97 mg/l, respectively.

The bar and liquid soaps composed of multiple fatty acid salts demonstrated 158 and 1,003 mg/l for 7d-LC$_{50}$, 158 and 902 mg/l for 14d-LC$_{50}$ (EC$_{50}$), and 101 and 284 mg/l for NOEC, respectively (Table 2).

### 2.3 Biodegradability tests

**2.3.1 OECD 301A.** C18:1Na and C18:1K demonstrated high biodegradability of 87.2 and 90.5%, respectively, in the OECD 301A test. When the bar and liquid soap were tested, the biodegradability was 60.6 and 67.6%, respectively (Table 3).

**2.3.2 OECD 301C.** C18:1Na and C18:1K demonstrated high biodegradability of 87.0 and 88.0%, respectively, in the OECD 301C test. When the bar and liquid soaps were tested, the biodegradability was 87.0 and 105%, respectively (Table 3). SDB merely exhibited −3%.

### 2.4. Human keratinocyte cytotoxic assay

To assess the cytotoxicity, the LDH leakage was measured, and cell viability was determined using the water-soluble tetrazolium (WST) assay.

**2.4.2. LDH assay and cell viability.** For the LDH leakage assay, after 5 min of treatment, NHEK-Ad cells treated with C12K demonstrated LDH leakage similar to the control value and significantly greater leakage only at higher concentrations of 31.2 mM. Similarly, LDH leakage treated with C18:1K increased significantly at 3.12 and 31.2 mM. However, the cells treated with SLS showed significantly greater leakage than the control value at lower concentrations, around 0.0312 mM (p < 0.05) (Fig 3).

**Table 3. Biodegradability of fatty acid salts, soaps and synthetic detergent.**

| No. | Sample | OECD 301A | OECD 301C | |
|---|---|---|---|---|
| | | Biodegradation degree (%) | Biodegradation degree (%) | Half life (day) |
| 1 | C18:1Na | 87.2 | 87.0 | 6.737 |
| 2 | C18:1K | 90.5 | 88.0 | 6.194 |
| 3 | Bar soap | 60.6 | 87.0 | 6.888 |
| 4 | Liquid soap | 67.6 | 105 | 1.514 |
| 5 | SDB | – | −3 | – |

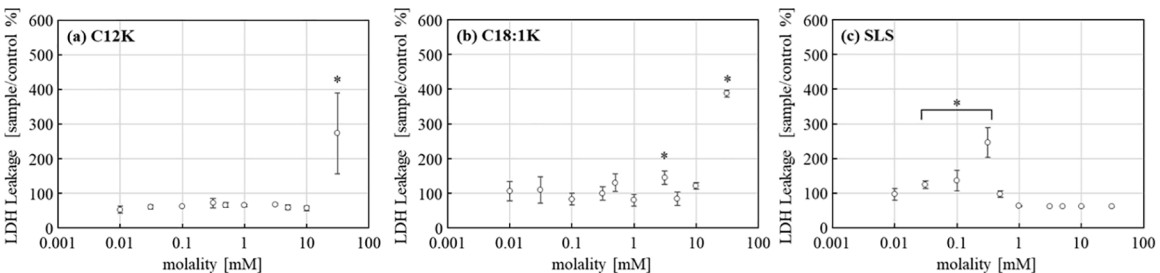

**Fig 3. LDH leakage of NHEK-Ad cells treated with (a) C12K, (b) C18:1K or (c) SLS for 5 min.** The cells treated with HBSS (+) were used as the control. The percentage of LDH leakage was calculated relative to the control value. Results are expressed as mean ± standard deviation (SD) values (*n* = 6). *Significantly larger than the control value (p < 0.05).

For cell viability, after 5 min of treatment with the test solutions, the cell viability of NHEK-Ad cells was measured. The cells treated with C12K or C18:1K showed cell viability similar to the control value at low concentrations but had decreased viability at 10 mM for C12K and 5 mM for C18:1K and higher concentrations. However, the cells treated with SLS showed decreased viability at 0.312 mM. The cells treated with C12K or C18:1K exhibited significantly greater viability than those treated with SLS at concentrations of 0.312 to 5 mM ($p < 0.05$). The $IC_{50}$ values of each solution for NHEK-Ad cells calculated from cell viability were 7.82, 7.56, and 0.604 mM for C12K, C18:1K, and SLS, respectively (Fig 4).

### 2.5. Questionnaires for clinical use

The mean ± SD of subjective questionnaire outcomes are 3.9 ± 0.6, 4.0 ± 0.1, 3.0 ± 0.9, 3.7 ± 0.6, and 3.5 ± 0.7 for the values of "itchy," "sticky," "moist," "dry," and "tight," respectively (Fig 5). Each value is higher, reflecting the nature of the natural soap. In historical comparisons to the healthy volunteers' values in handwashing, each value was 3.9 ± 0.3, 3.0 ± 0.9, 3.3 ± 0.7, 3.8 ± 0.4, and 3.9 ± 0.2, for the values of "itchy," "sticky," "moist," "dry," and "tight," respectively in Period I and 3.9 ± 0.3, 3.1 ± 0.7, 3.3 ± 0.7, 3.9 ± 0.4, and 3.9 ± 0.3, for the values of "itchy," "sticky," "moist," "dry," and "tight," respectively in Period II [6].

## Discussion

Soap is a salt of fatty acid produced by the saponification of animal fat, such as tallow and lard (pork fat), which contains fatty acids like the saturated palmitic and stearic acids, and the unsaturated fatty acid oleic acid. It can also be made from

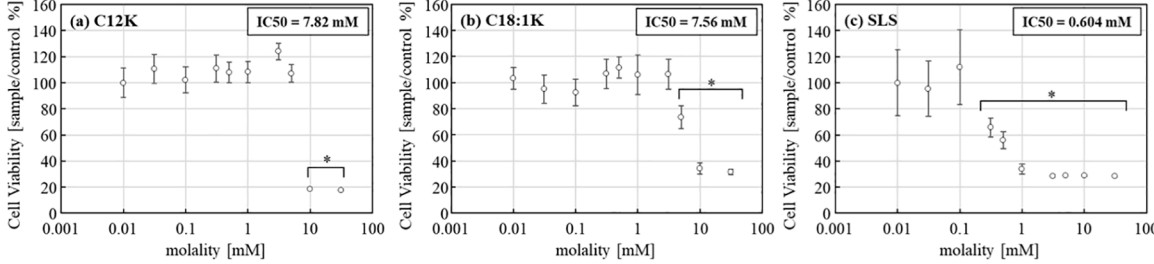

**Fig 4. Viability of NHEK-Ad cells treated with (a) C12K, (b) C18:1K or (c) SLS for 5 min.** The cells treated with HBSS (+) were used as the control. The percentage of cell viability was calculated relative to the control value. Results are expressed as mean ± SD values ($n = 6$). *Significantly lower viability than the control value ($p < 0.05$).

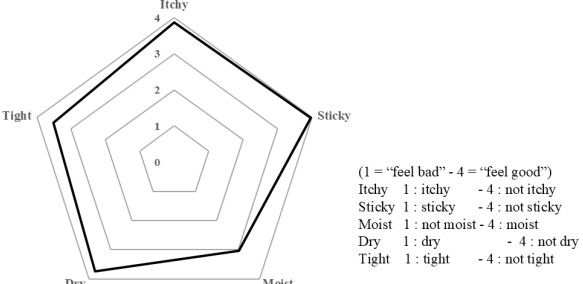

**Fig 5. Results of Questionnaires on the feelings after using Natural Soap.** The mean ± SD of subjective questionnaire outcomes are 3.9 ± 0.6, 4.0 ± 0.1, 3.0 ± 0.9, 3.7 ± 0.6, and 3.5 ± 0.7 for the values of "itchy," "sticky," "moist," "dry," and "tight," respectively.

vegetable fats, such as olive oil, palm oil, and coconut oil, in reaction with a strong base such as sodium hydroxide or potassium hydroxide. The most common fatty acid chain lengths are typically in the C12–C18 range [29], and our experiments were performed with these different alkyl chain lengths. In this study, bar soap is a mixture of sodium salts of fatty acids with an alkyl chain length of C12–C18, glycerin, and water, whereas liquid soap is a mixture of potassium salts of fatty acids with the same alkyl chain length, glycerin, and water. Notably, glycerin is not listed in the Pollutant Release and Transfer Register (PRTR) in Europe [30] or Japan (https://www.meti.go.jp/policy/chemical_management/english/files/SDSsystem.pdf; https://www.env.go.jp/en/chemi/prtr/substances/list.html).

Syndets are detergents synthesized from fats, petrochemicals, or oil-based products (i.e., oleochemicals) and alkali metals through chemical processes other than saponification, notably sulfonation. Anionic surfactants, such as SLS and linear alkylbenzene sulfonate (LAS, e.g., SDB), are often used in synthetic detergents. LAS is reported to be hazardous in aquatic environments [31], and SLS causes skin irritation by disrupting the skin barrier function [32]. A two-week exposure experiment was conducted to investigate the toxicity of LAS on five submerged macrophytes (four native and one exotic), focusing on their growth and physiological responses. The results showed that lower concentrations of LAS (<5 mg/L) slightly stimulated the growth of submerged macrophytes, while higher concentrations inhibited it. Increasing LAS levels led to a decrease in chlorophyll content, an increase in malondialdehyde content and peroxidase activity, and an initial rise in superoxide and catalase activities before a subsequent decline [33].

Gray water constitutes a significant fraction—50% to 80%—of total wastewater and has a cloudy appearance. It occupies an intermediate state between freshwater or drinking water and sewage. Recycling and reusing gray water are receiving increased attention due to its low concentrations of pathogenic microorganisms and nitrogen sources, despite its high phosphorus content.

One of the most problematic compounds in gray water is anionic surfactants, including LAS and branched alkylbenzene sulfonates (BAS), which are commonly used as ingredients in soaps and detergents.

A study investigating the molecular details underlying the biodegradation pattern of certain alkylbenzene sulfonates provided key insights. Two hydrogen bonds, essential for anchoring surfactants to the monooxygenase active site involved in the initial biodegradation step, were identified. These bonds determine how surfactants position themselves within the enzyme's hydrophobic pocket, thereby influencing the biodegradation rate in a structurally dependent manner.

These findings pave the way for the intelligent design of anionic surfactants that minimize their environmental impact in water and soil. To the best of our knowledge, this study represents a novel contribution toward the development of environmentally friendly surfactants with an enhanced likelihood of complete biodegradation and mineralization [34]. Further studies using natural soap, which is similar to anionic surfactants in gray water, will help clarify its environmental effects.

Algal 72h-EC$_{50}$ values were higher for all natural soap fatty acid salts compared to those of syndets (SDB and SLS). Longer alkyl chain lengths of the fatty acids demonstrated better performance, which may explain why both soaps have higher values than syndets.

Algal NOEC values at 72 h were higher for most natural soap fatty acid salts, except for C12K and C18K, compared to syndets. Both natural soaps were also less toxic. SDB was less toxic than SLS based on the algal EC$_{50}$ and NOEC values.

Crustacean 48h-EC$_{50}$ values were higher for the fatty acid salts, except for C18K, compared to those of syndets. All alkyl chain lengths of the fatty acids demonstrated better performance than syndets, with both soaps showing higher values than syndets.

Crustacean NOEC values for bar soap at 21 days were higher than those for C18:1Na and C18:1K.

Fish LC$_{50}$ values at 7 and 14 days were over 400 mg/l for C18:1Na and C18:1K. Bar soap and liquid soap exhibited LC$_{50}$ values of around 160 mg/l and 1,000 mg/l, respectively. Fish NOEC values at 14 days were around 100 mg/l for C18:1Na, C18:1K, and bar soap, while liquid soap showed 284 mg/l. In historical comparisons, fatty acid salts have been shown to be more effective than syndets [35–40]. In terms of biodegradability, the OECD 301A test is used to evaluate the

aerobic biodegradation potential of a fluid mixture by monitoring the removal of DOC from an aqueous solution by activated sludge and water microbial consortia. This is done for two substrate concentrations and four salinities. The OECD 301 test is a water-only system (Ministry of International Trade and Industry, Japan), and additional screening tests, such as the water-sediment system and soil screening test, have been developed [41] and further evolved into the OECD 301 test. The OECD 301 test evaluates ready biodegradability and allows for the addition of inert support or emulsifying agents, and/or modification of the test medium volume [42].

The pass level for the 28-day test duration in the OECD 301 test is 60% for BOD/ThOD (OECD 301C) and 70% for DOC removal (OECD 301A) [43]. Our data on the natural soap components of C18:1Na and C18:1K meet both criteria, and the bar soap and liquid soap passed the BOD/ThOD (OECD 301C) values. In contrast, the syndet, SDB, showed only −3% in the OECD 301C test, indicating that it is hardly biodegradable. Antiseptic synthetic hand soap often contains chloroxylenol (CHL), which serves as an alternative to triclosan and triclocarban—both of which have recently been banned in some countries. The more widespread use of CHL may significantly increase its presence and concentration in aquatic environments, potentially leading to ecological risks. Wastewater treatment plants (WWTPs) may act as point sources of CHL in natural environments due to its extensive discharge through urban waste streams.

Bioreactor operations and batch experiments were conducted to investigate the fate and effects of CHL and to elucidate the mechanisms underlying its degradation at various concentrations, ranging from environmentally relevant levels to high levels (0.5–5 mg L$^{-1}$). Bioreactors partially removed CHL (44–87%) primarily through biological processes. Microbial association networks, constructed using 16S rRNA gene sequencing data, revealed selective enrichment and a correlation between *Sphingobium* and CHL, suggesting its involvement in the biological breakdown of CHL through dehalogenation and ring hydroxylation pathways.

These findings provide valuable insights into the behavior and effects of CHL in activated sludge communities and offer important information for the sustainable management of CHL, which may become an emerging issue in the urban water cycle [44].

In the cytotoxicity assay, measured by LDH release after 48 h culture and 5 min exposure, both natural soap fatty acid salts, C12K and C18:1K, demonstrated significantly lower toxicity than the synthetic surfactant, SLS, at 1,000-fold and 100-fold, respectively. Cell viability, measured by the WST assay, indicated that C12K and C18:1K were 32-fold and 16-fold more viable than SLS. The IC$_{50}$ values were 7.82, 7.56, and 0.604 mM for C12K, C18:1K, and SLS, respectively.

The results of the questionnaires for clinical subjects were almost identical to those of the healthy volunteers using waterless methods with natural soap, except for the "sticky" category. The clinical subjects rated it 4.0 ± 0.1 in this study, compared to the handwashing volunteers' values of 3.0 ± 0.9 and 3.1 ± 0.7 in Period I and Period II, respectively [6]. This may reflect that barrier function-damaged skin may lead to more stickiness of the natural soap, which could also be favorable for clinical use.

## Conclusion

As the consumption of soaps and detergents has increased, concerns have arisen regarding their environmental effects and direct impacts on the human body. This study compares natural soap compounds, consisting of fatty acid salts, with synthetic detergents such as SDB and SLS.

Since little is known about the environmental impact of natural soap according to the environmental assays recommended by the OECD, as well as human keratinocyte assays for toxicity and biodegradability, these methods were utilized. The major components of natural soap were found to be less toxic and more biodegradable in aquatic environments—assessed using algae, crustaceans, and fish—compared to synthetic detergents. Additionally, in the human keratinocyte assay, natural soap compounds were significantly less toxic and demonstrated higher cell viability than SLS. The IC$_{50}$ obtained from the viability assay revealed values of 7.82 mM for C12K, 7.56 mM for C18:1K, and 0.604 mM for SLS.

Therefore, natural soap appears to be a valuable alternative due to its lower toxicity, greater biodegradability in aquatic environments, enhanced safety for human cells, and potential efficiency in clinical applications, unlike petroleum-based ingredients, which may require multiple steps to ensure safety.

## Acknowledgments

The authors appreciate Prof. Kohji Nakazawa, Department of Life and Environmental Engineering, The University of Kitakyushu, Kitakyushu 808−0135, Japan, for assisting the human keratinocyte assays; Kyushu Environmental Evaluation Association, Fukuoka, Japan, for assaying OECD 201, 202, 204, 301A; IDEA Consultations, Inc., Tokyo, Japan for assaying OECD 211; Hodogaya Contract Laboratory Co., Ltd., Ibaraki, Japan for assaying OECD 301C.

## Author contributions

**Conceptualization:** Sadanori Akita.

**Formal analysis:** Akihiro Masunaga.

**Funding acquisition:** Hayato Morita.

**Investigation:** Takahide Kanyama.

**Methodology:** Takayoshi Kawahara.

**Project administration:** Sadanori Akita.

**Resources:** Hayato Morita.

**Supervision:** Sadanori Akita.

**Validation:** Takayoshi Kawahara, Sadanori Akita.

**Visualization:** Akihiro Masunaga.

**Writing – original draft:** Takahide Kanyama, Akihiro Masunaga.

**Writing – review & editing:** Sadanori Akita.

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
