## [Decision Letter · Decision Letter 0]

Dear Dr. Akita,

Thank you for submitting your manuscript to PLOS ONE. After careful consideration, we feel that it has merit but does not fully meet PLOS ONE’s publication criteria as it currently stands. Therefore, we invite you to submit a revised version of the manuscript that addresses the points raised during the review process.

We look forward to receiving your revised manuscript.

Kind regards,

Muammar Qadafi

Academic Editor

PLOS ONE

Journal Requirements:

“This study was partly supported by non-restricted funds from the Shabondama Soap CO., Ltd.”

“This study was partly supported by non-restricted funds from the Shabondama Soap CO., Ltd. “

6. Thank you for stating the following in your Competing Interests section: “no”

7. Please provide a complete Data Availability Statement in the submission form, ensuring you include all necessary access information or a reason for why you are unable to make your data freely accessible. If your research concerns only data provided within your submission, please write "All data are in the manuscript and/or supporting information files" as your Data Availability Statement.

8. Please ensure that you refer to Figure 5 in your text as, if accepted, production will need this reference to link the reader to the figure.

9. Please include your tables as part of your main manuscript and remove the individual files. Please note that supplementary tables (should remain/ be uploaded) as separate ""supporting information"" files

Reviewers' comments:

Reviewer's Responses to Questions

**Comments to the Author**

1. Is the manuscript technically sound, and do the data support the conclusions?

Reviewer #1: Yes

Reviewer #2: Partly

2. Has the statistical analysis been performed appropriately and rigorously?

Reviewer #1: Yes

Reviewer #2: Yes

3. Have the authors made all data underlying the findings in their manuscript fully available?

Reviewer #1: Yes

Reviewer #2: Yes

4. Is the manuscript presented in an intelligible fashion and written in standard English?

Reviewer #1: Yes

Reviewer #2: Yes

Reviewer #1: Natural Soap is Less Toxic and More Biodegradable in Aquatic Organisms and Human Skin Cells than Synthetic Detergents and clinicallyefficient (suggest revise due to minor spelling and sentencing errors)

Abstract: well written

Introduction: well written

Materials & methods: Generally well written, suggest to provide abbreviation list. slight proofreading needed for the statistics section.

Results:

* Mostly well written,

* Need a compiled list of abbreviations,

* Once a term is abbreviated (i.e., sodium oleate (C18:1Na)), suggest to keep using the abbreviated version, instead of constantly keeping them in brackets.

* Suggest to review some sentences for grammatical errors.

Images used are a bit blurry, suggest to substitute with at least 300dpi images.

* Sections 2.2.1/2.2.2/2.2.3 : Result write-up should refer to the respective table immediately at the end of sentences or at the end of the paragraph, instead of at the end of the section.

* Section 2.5: The results provided should refer to a graph/table instead of to a bibliographic reference [6]

Discussion

- Generally well-written but had a heavy focus on results that have been presented well in the results section

- Suggest to make comparisons from the literature on similar studies and their findings.

- Suggest to add relevant environmental, economical, community & health impacts of the research

- Suggest to include the study strength and limitation as well as suggestions for future improvement etc

Reference:

Inadequate references for an original paper. Suggest to expand the discussion section which will eventually expand the numbers of references.

Conclusion:

- a good study and a generally well written manuscript, but needs improvement & expansion in the discussion to strengthen the manuscript further

Reviewer #2: 1. Research Title: The title "Natural Soap is Less Toxic and More Biodegradable in Aquatic Organisms and Human Skin Cells than Synthetic Detergents and Clinically Efficient" suggests a conclusion that natural soap is indeed less toxic and more biodegradable, while the aim of the study is to explore these aspects. Would it be better to use a title such as: “Comparative Study on the Toxicity and Biodegradability of Natural Soap versus Synthetic Detergents in …” to better align with the study's objectives?

2. [Introduction]: This study has outlined various aspects of the differences between natural and synthetic soaps, including toxicity, biodegradability, effects on aquatic organisms, and impacts on human health and the environment. However, the research gap has not been explicitly stated. It would be beneficial to include a statement explaining what has not been thoroughly studied and why it is important.

3. [Methodology]: In section 1.2.1, please explain the basis for determining the number of replicates, specifically the 3 replicates for treatments and 6 replicates for the control group.

4. [Methodology]: In section 1.4.3, concerning the LC50 equation [mM], is the use of log-log (double logarithmic) correct, or is it just a single logarithm? Please clarify.

5. [Methodology]: In section 1.5, it is stated that the inclusion and exclusion criteria are in Table 3, but I could not find Table 3 in your submission. Please verify and include it.

6. [Methodology]: In section 1.5, it is mentioned that there are 50 study participants, including 4 other conditions. Please provide an explanation of these other conditions, as they represent variables in the study that need to be detailed.

7. [Methodology]: What type of statistics did you use in the analysis? Did you perform a test for differences in values? If so, please clarify which values were tested for differences.

8. [Results]: I also could not find Table 1 and Table 2 in your submission, making it impossible for me to check the contents of these tables. Kindly include the tables in the submission.

9. [Results]: This study primarily focuses on assessing the toxicity of natural soap components on aquatic organisms, its biodegradability, and its cytotoxic effects on human keratinocytes. However, does this natural soap also provide superior antibacterial effectiveness compared to synthetic soap, given that the main function of soap is antibacterial, particularly in light of the COVID-19 pandemic background? It would be useful to include literature studies supporting the sustainability of your findings.

10. [Conclusion]: I could not find a conclusion section in this submission. Please add a conclusion that includes the main findings and their implications.

11. Figure 5: The title is simply "Research of Questionnaires." The title of Figure 5 should be more descriptive, such as "Results of Questionnaire on the Effects of Natural Soap and Synthetic Detergents," and include explanations and units for each value from 1 to 4 to improve clarity.

12. I could not find an explanation of the abbreviation HBSS in the article. It would be helpful to define HBSS when it is first mentioned in the article, followed by the abbreviation in parentheses.

Thank you,

February 17, 2025.

**Do you want your identity to be public for this peer review?** For information about this choice, including consent withdrawal, please see our Privacy Policy

Reviewer #1: **Yes: ** INTAN SUHANA ZULKAFLI

Reviewer #2: No

---

## [Author Response · Author response to Decision Letter 1]

19 Apr 2025

Thank you and the authors wish the revision is satisfactory for publication.

---

## [Decision Letter · Decision Letter 1]

Natural soap is clinically effective and less toxic and more biodegradable in aquatic organisms and human skin cells than synthetic detergents

PONE-D-25-01183R1

Dear Dr. Akita,

We’re pleased to inform you that your manuscript has been judged scientifically suitable for publication and will be formally accepted for publication once it meets all outstanding technical requirements.

Kind regards,

Muammar Qadafi

Academic Editor

PLOS ONE

Additional Editor Comments (optional):

Comments from Reviewer 2 should be addressed during the final proof.

Reviewers' comments:

Reviewer's Responses to Questions

**Comments to the Author**

Reviewer #2: All comments have been addressed

2. Is the manuscript technically sound, and do the data support the conclusions?

Reviewer #2: Yes

3. Has the statistical analysis been performed appropriately and rigorously?

Reviewer #2: Yes

4. Have the authors made all data underlying the findings in their manuscript fully available?

Reviewer #2: Yes

5. Is the manuscript presented in an intelligible fashion and written in standard English?

Reviewer #2: Yes

Reviewer #2: Thank you for the response provided. All comments have been addressed to improve the quality of the article. However, I have noticed that there are still several sentences with excessive spacing, making the text appear untidy. Kindly review and revise the entire article to ensure proper formatting.

In addition, we have also reviewed Table 3, particularly point 5, where a percentage value of "-3" is shown. Please recheck whether this value is indeed a negative number or if it is a typographical error.

**Do you want your identity to be public for this peer review?** For information about this choice, including consent withdrawal, please see our Privacy Policy

Reviewer #2: No

---

## [Editor Report · Acceptance letter]

PONE-D-25-01183R1

PLOS ONE

Dear Dr. Akita,

I'm pleased to inform you that your manuscript has been deemed suitable for publication in PLOS ONE. Congratulations! Your manuscript is now being handed over to our production team.

Kind regards,

on behalf of

Dr. Muammar Qadafi

Academic Editor

PLOS ONE